# Reviving Pagan Spirituality: A Manifesto

Keith Parsons 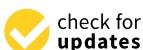

College of Human Sciences and Humanities, University of Houston-Clear Lake, Houston, TX 77058, USA;
parsons@uhcl.edu

**Abstract:** Numerous contemporary neopagan movements are attempts to revive or reconstruct ancient religious belief and practice. For instance, the worship of the ancient Norse gods has been restored to Iceland by the Asatru Fellowship. In this essay, I defend neopagan movements against the charge that ancient spirituality cannot be recovered in identifiable form. I note that today's dominant religions, such as Christianity, also face questions of the continuity of identity and argue that if such problems are tractable for current religions, then, in principle, they are resolvable for neopagans. I further argue that there are three broad themes of spirituality that are identifiable in ancient pagan religion, and that these are permanent possibilities recoverable by modern people. I also defend the relevance and importance of these themes.

**Keywords:** neopagan; pagan; Asatru; soteriology; Axial Age; mythology; Norse; classical Greece; John Hick; Paul Tillich; Mircea Eliade; sacred space

In the year 1000 C.E., Iceland became Christian. Now, after a hiatus of over a thousand years, a temple to the Old Gods is being built in Reykjavik. In 1973, Ásatrú, the worship of the Norse gods and goddesses, was officially recognized as a practiced religion in Iceland, meaning that the Ásatrú Fellowship (*Ásatrúarfelagið*) qualified to receive state support (Paxson 2006, p. xiii). Icelandic Ásatrú is hardly an isolated phenomenon; it is one of many religious movements that seek to revive or reconstruct ancient religions:

> These include Celtic traditions, among them the different kinds of Druids; the Hellenic traditions, which draw from ancient Greece; the Kemetics, who base their practice on the religion of Egypt; Baltic traditionalists, who have revived their native religions in their newly independent nations; and the religions of the Germanic peoples in Scandinavia, on the Continent, and in England. (Paxson 2006, p. xxii)

Each of these movements represents an effort to restore a kind of spirituality that is not merely pre-Christian but pre-Axial Age.[1] The distinctive features of Judaism (and, consequently, Christianity and Islam), Hinduism, Buddhism, Confucianism, and Taoism took shape during this "Axial" period. We may call "pre-axial" the "national" religions of ancient Egypt, Babylon, Greece, and the Norse/Teutonic regions of northern Europe, as well as the animistic and shamanistic religions of preliterate peoples.

I could refer to these ancient religions as "archaic," but I will call them "pagan," though that was originally a term of abuse. I call them "pagan" because that is the term that is commonly used by those who today identify with those ancient religions, and who often refer to themselves and their practice as "pagan" or "neopagan."

However, is it even possible to revive an ancient religion? Can we engage in the authentic spirituality of people who lived millennia ago in very different material and intellectual conditions, or will we only be fooling ourselves? Put bluntly, will we be indulging only in a kind of religious cosplay, pretending to be something we can never really be, like those who dress up as Klingons or Jedi at fantasy conventions? I will consider Icelandic Ásatrú as a case study and respond to Michael Strmiska's argument that Icelandic paganism cannot be reborn (Strmiska 2000).

To address Strmiska's argument we will have to ask just what it means to say that a religion is "reborn," "revived," or perhaps "reconstructed," and this will require an analysis of what qualifies as identity versus innovation with respect to religious belief and practice. My conclusion will be that certain forms of spirituality transcend the limitations of time and place and represent permanent possibilities of relating to the sacred. If ancient people were aware of such possibilities, there is in principle no reason why they cannot be rediscovered. I argue that it is apt to speak of the "revival" of ancient religions but not a literal rebirth. That is, that core ancient pagan beliefs, practices, and modes of spirituality can be identified and legitimately appropriated by today's neopagans.

## 1. Can Nordic Paganism Be Reborn?

In his article "Ásatrú in Iceland: The Rebirth of Nordic Paganism?" Michael Strmiska questions the possibility of the revival of Icelandic paganism:

> . . . I will attempt to show that however much this religion attempts to revive elements of the pre-modern Pagan past, it is in fact a quite postmodern movement. The self-understanding of the religion's beliefs and conception of the sacred are riddled with uncertainty and historical confusion. Within the movement there are many possibilities of understanding and experience and the believers lack the means to resolve these ambiguities in a clear and compelling manner—a dilemma entirely reflective of our current historical period rather than of the era to which the movement hearkens back (Strmiska 2000, p. 106).

Strmiska is not a hostile or ideological critic of Icelandic Ásatrú. On the contrary, he has firsthand experience of the ceremonies of the Ásatrú Fellowship, and frequently expresses admiration for their beauty. His chief objection arises from his observation of irresolvable conflict about the appropriate nature of worship and ceremony:

> One faction . . . is eager to have more . . . exuberant, participatory activities, and to move in a direction of more ecstatic and sensually exciting experience. Another camp . . . prefer more staid and dignified procedures. For them, a cheerful evening of shared food, drink, and heartfelt recitations of poetry in a consecrated setting is sufficient (p. 123).

> These divisions became deep enough to lead to the departure of some members from the Fellowship (p. 123).

> Strmiska comments:

> This dispute points to a fundamental problem which goes to the very heart of the Neopagan enterprise. By [High Priest] Jormundur's own admission, the surviving texts and other related materials concerning the original Norse Pagan religion are too fragmentary and incomplete to provide a definitive basis for all Ásatrú rituals and pursuits, and must be judiciously supplemented by ideas and practices improvised in the present or borrowed from other sources. This is where the dilemma arises if Ásatrú or any other Neopagan group, goes too far from its original core traditions, the sacred forms of the cherished Pagan past, it loses its claim to authenticity. But if it is so slavish and bookish in its fealty to ancient lore that it excludes new possibilities of encountering or conceptualizing the sacred, it closes itself off from spiritual vitality (p. 128).

Yet such conflicts relating to differences in belief and practice have afflicted all religions. Christianity is a prime example. Intractable disagreements over Christian doctrine and worship have embroiled the Christian Church from the earliest times, as evidenced by the letters of St. Paul. Much of the subsequent history of the Church is a story of schism, conflict, faction, and the proliferation of "heresies." Repeated attempts to impose unity by appeals to authority or force were never wholly successful. Efforts by Christians of later ages to identify an "original" *kerygma* or to reconstruct the historical Jesus have been "riddled with uncertainty and historical confusion." There are (and always have been)

many Christianities, "many possibilities of understanding and experience," and it may be fairly added that "the believers lack the means to resolve these ambiguities in a clear and compelling manner." As for styles of worship, current Christianity evinces an enormous diversity, from Pentecostal ecstasies to the most reserved high church liturgies. Mutatis mutandis, the same sort of thing may be said about other major religions. It appears, then, that a defender of Ásatrú would at least have *tu quoque* replies.

All religions also face the dilemma of identity versus innovation. What, after all, is continuity in religion? Continuity of religious identity is not the possession of an immutable essence. The various fundamentalist and ultra-orthodox movements in all religions are only deluding themselves in thinking that they alone possess such an essence pure and unsullied. *Any* reading of an ancient text, no matter how strongly one is committed to expounding "original intent," is inevitably an interpretation that imports assumptions and modes of thought reflective of one's own intellectual and cultural milieu. Further, religions constantly reinvent themselves, and necessarily so. Such continuity as they possess is a matter of family resemblances in the Wittgensteinian sense—connections maintained, broken, reestablished, modified, and projected across time and place. What would a first-century Christian think if transplanted into one of today's megachurches? Yet, if Christian identity is possible across the upheavals of 2000 years, considerable latitude must be granted to the members of the Ásatrú Fellowship in their claim of identity with ancient Norse religion.

To assess that claim we need a clarification of terminology. Strmiska uses the terms "rebirth," "revival," and "reconstruction" without clearly distinguishing their meanings. I think we should. Let us say that a religion is "reborn" if it is brought back in its exact original form, with all of the beliefs and practices of its ancient adherents. To say that a religion is "revived", on the other hand, should mean that core elements of ancient spirituality have been found and reclaimed in identifiable form, recognizing, for reasons given in the previous paragraph, that our appropriation may differ significantly from its ancient instantiations. A "reconstruction" of an ancient religion would be its transformation into a self-consciously modern religion, with elements "inspired by" ancient traditions and others improvised or borrowed from other traditions. Which term best describes what modern pagans such as the Ásatrú Fellowship are doing?

I think it is obvious that it is neither possible nor desirable to bring back an ancient religion in its precisely original form. For one thing, it is doubtful that ancient religions had a single, unitary form. Rather, there were probably considerable differences in practice from one time or locale to another.[2] Further, our evidence for ancient beliefs and practices is usually too skimpy to be sure that we have recovered in toto any version of an ancient religion. Finally, some aspects of ancient religions, such as human sacrifice, are obviously not desirable to recover.

I believe that Strmiska would agree that an ancient religion can be reconstructed, that is, turned into a self-consciously modern religion that is motivated and inspired by an appreciation of ancient traditions. Is it appropriate to say that Icelandic Ásatrú has "revived" and not merely "reconstructed" the ancient Norse religion? The answer depends upon our assessment of how successfully they have addressed the identity/innovation dilemma. Clearly, modern followers of Icelandic Ásatrú believe differently from the ancients. As Strmiska notes, hardly any of them now literally believes in the Norse deities. It is safe to say that hardly anyone now believes that Odin is riding about on his eight-legged horse or that Thor is pummeling giants with his mighty hammer. The attitude of modern-day followers is much like that of liberal Christians towards a literalistic reading of the Adam and Eve story. While such narratives are taken as having allegorical, symbolic, or metaphorical significance, they are not regarded as actual history. Of course, fundamentalists regard such liberals as "not real Christians," but such judgments can only be based on question-begging criteria.

A deeper qualm about the recovery of an ancient spirituality is this: Religions have a particularity of time and place; they are both the cause and effect of the culture of the

societies in which they are embedded. Cultural identity necessarily includes religion among its essential components; that is, a society, past or present, is just not identifiable without reference to its religion. Is this relationship between culture and religion symmetrical? If culture cannot be understood without reference to religion, is religion likewise inextricable from its ambient culture? Must we assume that religion is always and only an idiosyncratic and strictly culture-bound enterprise, expressive only of the mores of a delimited time and place?

Some of the greatest religious thinkers, including Rudolf Otto, Mircea Eliade, Paul Tillich, and John Hick have disputed such an astringent view and have argued that basic elements of religiosity are transcultural and transhistorical. Tillich, for instance, famously identified religion with the "depth" dimension and ultimate concern:

> [Religion] is at home everywhere, namely, in the depth of all functions of man's religious life. Religion is the dimension of depth in all of them. Religion is the aspect of depth in the totality of the human spirit. What does the metaphor *depth* mean? It means that the religious aspect points to that which is ultimate, infinite, unconditional in man's spiritual life. Religion, in the largest and most basic sense of the word, is ultimate concern. And ultimate concern is manifest in all creative functions of the human spirit (Tillich 1959, pp. 7–8).

Otto identified a primal sense of the sacred, the holy, or the numinous (his own word) as the core of all religions: "There is no religion in which it does not live as the real innermost core, and without it, no religion would be worthy of the name." (Otto 1958, p. 6). Hick's magisterial *An Interpretation of Religion* defends a pluralistic view that regards all religions as human responses to the transcendent—what he calls "The Real"—a reality that is encountered by all, but which cannot be fully encompassed by human concepts.

There therefore appears to be no basis for dismissing a priori the claim that core elements of ancient religious belief and practice can be identified and appropriated by latter-day adherents. Yet is it really possible to reclaim a lost religiosity? In general, we have a variety of means of accessing ancient religious belief and practice. For the Norse/Teutonic religion, for instance, there are rich textual sources such as the Norse myths as recorded in the *Eddas*, Icelandic sagas, and Germanic heroic poems. There is also the evidence supplied by artifacts and archaeology, such as runestones, carvings, and gravesites. Of course, we would like to know more, but through these sources we can discern a cosmology and vibrant mythology that tells us much about the values, worldview, and spirituality of the ancient peoples of northern Europe. Modern followers of Ásatrú must supplement these sources with imaginative reconstructions of ancient practice, but, as we say, many other religions must do the same.

I claim that, considering ancient pagan traditions in general, and not just their Norse/Teutonic versions, we may identify three very broad themes of pagan spirituality, themes that can be differently appropriated by worshippers today. I defend each of these claims in separate sections below.

(1)    Religion is not about salvation or liberation from a putatively corrupt, fallen, or debased reality. Rather, religion is about maintaining order, balance, and harmony in our relationships with the gods, with natural forces, and with each other. Life is not seen as in need of wholesale redemption or transformation, nor are believers expected to achieve a radical new self-understanding. The world is not in any sense to be abjured or escaped. On the contrary, religion affirms and celebrates the value of earthly life. Religious practices provide stability through life's vicissitudes and continuity through major transitions. Participation in ceremonies and rituals reinforces a sense of solidarity and identity within communities.

(2)    The divine is present, pervasive, and accessible, not sequestered in a distant realm. Access to the divine is not controlled by institutions or parceled out through authorized providers. The gods are there for everyone. Further, there is no sharp distinction between the sacred and the secular. Ritual and ceremony are important parts of the

religious life, but spiritual potencies permeate nature and the mundane so that there is no hard division between ordinary life and special religious occasions.

(3)   Myth is powerful. Religious identity is not established by creed, a declaration of required beliefs, but by sharing in a rich body of myth. Myth is a heterogenous set of narratives originating in the distant past and told and retold across the generations. Telling and hearing these narratives unites people with their past, affirms their basic values, and reinforces their membership in a community. Myth does not contrast with truth, but is defined by its function in conveying meaning and a sense of belonging and identity.

In the remainder of this essay, I will elaborate and defend these claims as elements of a viable neopagan religiosity.

## 2. Salvation versus Affirmation

Should religion be about liberation from the world or about living harmoniously in the world? Hick identifies a soteriological emphasis as distinguishing those religions that arose during and after the Axial Age. Of course, those religions do instruct their followers about the conduct of their quotidian lives, sometimes in excruciatingly minute detail. Yet themes of salvation, redemption, transcendence, liberation, escape, release, and personal transformation do play an essential role in the doctrines of the world's currently dominant religions, while being notably absent from pre-axial traditions. Rather, the emphasis of pre-axial religion was on providing guidance and meaning for individuals and the preservation of balance, cohesion, and harmony in society. Hick comments:

> Pre-axial religion has both psychological and sociological dimensions. Psychologically it is an attempt to make stable sense of life, and particularly of the basic realities of subsistence and propagation and the final boundaries of birth and death, within a meaning-bestowing framework of myth. This serves the social functions of preserving the unity of the tribe or people within a common world-view and at the same time of validating the community's claims upon the loyalty of its members. The underlying concern is conservative, a defense against chaos, meaninglessness and the breakdown of social cohesion. Religious activity is concerned to keep fragile human life on an even keel; but it is not concerned, as with post-axial religion, with its radical transformation (Hick 2004, p. 23).

With their vague beliefs about the afterlife and this-worldly emphasis, pre-axial religions had no notion of eschatology or salvific transformation to a higher level of being:

> The religious system functioned to renew or prolong the existing balance of good and evil and to ward off the possible disasters which always threatened. But it did not have in view any transformation of the human situation. There was no sense of a higher reality in relation to which a limitlessly better future is possible (Hick 2004, p. 28).

For pre-axial religion there was no idea that natural human life was radically defective or that the human personality required extensive overhaul. On the contrary, religion affirmed and reinforced the goods potentially enjoyable in earthly life. Of course, life had its ups and downs, and pain, sickness, and death always threatened, but the idea that life or human nature is comprehensively and fundamentally flawed was unknown.

For post-axial religions, on the other hand, motifs of the radically unsatisfactory nature of human life are central themes:

> They [post-axial religions] all recognize, first, that ordinary human existence is defective, unsatisfactory, lacking. For the Jew, we suffer from an innate inclination to evil, the *yetzer ha-ra*, and we live in a world in which evil forces have long been harassing God's chosen people. For the Christian, this is a "fallen" existence ruined by the primordial sin of our first ancestors. Inheriting their fault, or its consequences, we live in alienation from God, from ourselves, and from one

another. For the Muslim we human beings are weak and fallible and our life is commonly lived in *ghafala*, forgetfulness of God . . . For Hindus of all kinds, as also for the Jains and in modern times the Sikhs, the ordinary human condition is one of immersion in the relative illusoriness of *avidyā*, subject to the recurrent pains and sorrows of birth and death round which we are propelled by our karmic past. And for the Buddhist, the first Noble Truth is that all life involves *dukkha*, an "unsatisfactoriness" which includes pain, sorrow, and anxiety of every kind (Hick 2004, pp. 32–33).

Having identified the putative diseases, post-axial religions offer the putative cures:

The great post-axial traditions . . . exhibit in their different ways a soteriological structure which identifies the misery, unreality, triviality, and perversity of ordinary human life, affirms an ultimate unity of reality and value in which or in relation to which a limitlessly better quality of existence is possible, and shows the way to realize that radically better possibility. This may be by self-committing faith in Christ as one's lord and savior; or by the total submission to God which is *islam*; or by faithful obedience to the Torah; or by transcendence of the ego, with its self-centered desires and cravings, to attain *mokṣa* or Nirvana (Hick 2004, p. 36).

Tillich echoes these themes by talking about the "emergency character" of religion (Tillich 1959, p. 9), the emergency being "the tragic estrangement of man's spiritual life from its own ground and depth." (Tillich 1959, p. 8).

How would the intelligent pagan respond to such talk? I think he or she would ask, "What emergency? What estrangement? What limitlessly better existence?" Some years ago, in the Atlanta area, a local evangelical church printed up bumper stickers for display by its members reading "I found it!" A local Jewish congregation responded with stickers reading "We never lost it!" I think the pagan's reply to talk of alienation or estrangement would be similar. To the extent that modern humans experience angst, anomie, or alienation, the pagan would indeed consider such a condition as the result of a fall—the fall from paganism. Hick partially agrees:

The profound changes initiated during the axial age brought loss as well as gain. In pre-literate tribes life's hardships are to be endured and its joys communally celebrated in ways that largely unknown to us individualized men and women. In the archaic religions of the ancient Near East and of India there were an affirmation of life and a natural acceptance of death which have been largely lost since the discovery of sin and salvation, *avidyā* and illumination. Indeed, the axial age could be seen as the fall of humanity from a state of religious innocence (Hick 2004, p. 28).

I think our hypothetical pagan would reply "'Discovery?' No. 'Invention.'" Such a pagan might agree that modern humans are indeed alienated from nature, from their neighbors, and from themselves. All three forms of alienation are evident in the self-induced solipsism of our addiction to electronic devices. I will not launch into a neo-Luddite jeremiad, but I will just note the radical impoverishment implied by internet "friends" as opposed to real friends, by absorption in cyber worlds rather than the real one, and by a "social" media that serves largely to increase isolation, polarization, and tribalism. Neopagans would offer a cure in the form of reconnection, not repudiation, release, or redemption.

## 3. The Nearness of the Gods

For the classical Greeks, there was no clear demarcation between the secular and the sacred or between their religious and civic life. The great annual religious festivals were sponsored by the state, and participation in them was both a religious and a civic duty. In fact, the concept of "religion" as a distinct category did not exist:

The Greeks had no word for religion. Gods were thought to be everywhere, and religion was a part of everyday life: it was not divorced from mundane activities and therefore no word categorized it (Adkins and Adkins 1997, p. 284).

Greek religiosity thus retained features of preliterate religion as characterized by Hick:

Whereas in the thinking of modern technological people "the spiritual" is generally relegated to a margin of private fantasy or "faith," it seems that for pre-literate people it has always been a part of the world. The forests, hills, streams, rocks, sky are full of unseen beings and forces which have to be taken into account. There are local gods and spirits . . . who are to be variously worshipped. There are magical and ritual practices of many sorts. In all this there is no division between ordinary secular life and special religious moments but rather a single seamless fabric in which what the modern world sees as the "natural" is everywhere suffused with "supernatural" presence and meaning (Hick 2004, p. 24).

As Hick notes, for ancient people the natural world was full of gods, so divinity was as close as the local river or forest. Artists have evoked the sense of awe and dread that ancient people felt living within and wholly dependent upon of a vast landscape and its resident gods. Sibelius' symphonic poem *Tapiola* depicts the dark, brooding northern forest and Tapio, the mighty forest god of pagan Finns. Some places had a particularly sacred character, that is, they were places where the presence of the divine was particularly felt. Sacred natural places have always been important for Native American religion, and monuments such as Stonehenge show the efforts to which ancient people would sometimes go to construct their own sacred spaces.[3]

Not only were the gods near, you could also variously interact with them. In the scene that opens the *Iliad*, Achilles, furious at the greed and arrogance of Agamemnon, starts to draw his great sword, but he suddenly feels a tug on his hair. He turns to see the blazing gray eyes of Athena who warns him that Hera orders him to control his anger. Reluctantly, he obeys and lets his sword slip back into the scabbard. He verbally lashes Agamemnon and then departs to his tent to nurse his injured pride and to begin his famous sulk. One notable thing about this passage is that Achilles, though surprised, does not appear particularly startled by the theophany, and is certainly not overawed. Interactions between gods and humans occur often in the *Iliad.*

Everyday Greeks, and not merely Homeric heroes, were on familiar terms with their gods:

. . . they [the Greeks] did not think that the deities on whom they most depended were transcendent and far-removed. Rather, they were close at hand, as close as the hearth (Hestia), the *herma* or boundary stone in the street (Hermes), the shrine before the house, which was perhaps sacred to the Apollo of the Roads, the large jar in the storeroom sacred to Zeus Ktesios (guardian of the family possessions), and the courtyard watched over by Zeus Herkeios . . . All formal occasions required the invocation of a god or gods—marriage, for instance, or the reception of a newborn baby into the family circle, or at the death and burial of members of the family. Farming and other occupations could not be successfully pursued nor a journey on land or sea attempted without approval of the gods. The address to the gods on such occasions was simple and courteous but not servile, a natural, almost unreflective gesture of cooperation and community, not dominated by fear (Noss 1969, p. 55).

So, the Greek gods were near and familiar and present in the everyday world. For the ancient Greek, Zeus was a partner or patron, and the relationship was one of cooperation, not command and obedience. Zeus expected only courtesy and respect and was not jealous of partnerships with other gods. By contrast, the God of the Abrahamic religions is The Lord, who demands obedience, submission, and a commitment that is absolute and exclusive.

For ancient and modern followers of Norse paganism, the relationship with the gods is even closer:

> The old Viking concept of "friendship" with the gods characterizes the relationship many heathens have with their deities today. In *Eyrbyggjassaga,* Thorolf Mostur-Beard is described as "a close friend of Thor" . . . In *Gisli's Saga,* a man called Thorgrim was "a friend of Freyr" . . . Friendship provides a useful model for our relationship with the gods and goddesses and other wights. Like any other relationship, friendship with a god requires mutual respect and attention. We talk to our deities, share our food and drink with them, and quiet our minds so that we can hear what they have to say (Paxson 2006, pp. 98–99).

What would this idea of the nearness and accessibility of the gods mean to someone like the modern follower of Icelandic Ásatrú who no longer believes in the literal existence of the gods? I think it would mean this: That daily life is suffused with spiritual potencies, that is, that even in the midst of the most mundane activities, we can experience awe and wonder that is properly termed "religious" in its quality and intensity. What? When cutting grass, washing dishes, or stuck in traffic, we can have a frisson of the holy, a taste of the *mysterium tremendum*? We can. I have. Usually, we don't. The world is too much with us, as Wordsworth noted, and he envied the "pagan suckled in a creed outworn," who might

> Have sight of Proteus rising from the sea;

> Or hear old Triton blow his wreathèd horn.

Nature, music, art, acts of especial charity or kindness, and the love of human or animal can usher us into the precincts of the sacred—if we let them. As Hick noted, the archaic worshipper recognized the interpenetration of the sacred and the secular, the mundane and the transcendent. Such paganism was not a creed outworn but is wiser than we.

### 4. The Power of Myth

What is the function of myth? Eliade says that myth brings the sacred into the world, and explains both how and why the world exists:

> The myth reveals absolute sacrality, because it relates the creative activity of the gods, unveils the sacredness of their work. In other words, myth describes the various and sometimes dramatic irruptions of the sacred into the world . . . It is the irruption of the sacred into the world, an irruption narrated in the myths, that *establishes* the world as a reality. Every myth shows how a reality came into existence, whether it be the total reality, the cosmos, or only a fragment—an island, a species of plant, a human institution. To tell how things came into existence is to explain them and at the same time indirectly to answer another question: *Why* did they come into existence? The why is always implied in the how—for the simple reason that to tell *how* a thing was born is to reveal an irruption of the sacred into the world, and the sacred is the cause of all real existence (Eliade 1959, p. 97; emphasis in original).

In myth, the how and the why are united; for us today they are not. Myth explains in terms of the actions of personal agents, whereas such accounts were long ago barred from natural science. Explanations in the natural sciences are in terms of impersonal entities, processes, forces, and laws—and rightly so. Attempts to inject personal explanations into modern science result in absurdities such as "scientific creationism" and "intelligent design theory." On the other hand, if we take myth as myth, and do not turn it into a pseudoscience, how can we view myth as anything but quaint stories made irrelevant by science?

Maybe we just interpret away the mythical element. In the twentieth century, theologian Rudolf Bultmann argued that the mythical context of religion, if taken as an outdated cosmology, becomes a stumbling-block for modern persons, preventing them from confronting the true, radical message of the gospels. He therefore recommended that

Christian belief be "demythologized (Bultmann 1958)." For Bultmann, "demythologizing" the New Testament message is not a cut-and-paste job—like Thomas Jefferson's version of the Gospels with the miraculous bits excised—but a radical reinterpretation in the light of his identification of a "true" meaning. Drawing upon Heideggerian philosophy, Bultmann interpreted the gospel message as the confrontation of human beings with the choice between an authentic and inauthentic existence.

By contrast, the neopagan recommendation is to "re-mythologize." That is, that we should immerse ourselves in the great myths, determined to let them speak to us rather than to press them into our philosophical molds. Of course, we are not our ancient ancestors. As noted earlier, probably few people today think that Thor is slaying giants and that Odin is galloping on his eight-legged steed. However, the power of a story is not a function of its perceived factuality, but of the deep emotional resonance it has for us. Myth can be amusing, and one of the chief values of myth is that the tales are often so much fun. On the other hand, myth can be disturbing, even terrifying. Like music, myth can provoke many different feelings. The cognitive value of myths, the lessons we can learn from them, is not a product of philosophical analysis, but a sharpening, clarifying, or deepening of our intuitions and the enlivening of our imagination.

Consider the closing lines of James Weldon Johnson's magnificent version of the Biblical creation myth:

Up from the bed of the river
God scooped the clay;
And by the bank of the river
He kneeled Him down;
And there the great God Almighty
Who lit the sun and fixed it in the sky,
Who flung the stars to the most far corner of the night,
Who rounded the earth in the middle of His hand;
This Great God,
Like a mammy bending over her baby,
Kneeled down in the dust
Toiling over a lump of clay
Till He shaped it in His own image;

Then into it He blew the breath of life,
And man became a living soul.
Amen. Amen.

Myth, like all great literature, empowers us to transform our perceptions, to discover significance, and to *feel* truth and not merely to acknowledge it.

Is the transformative power of Johnson's poetry diminished by knowing astrophysics and evolutionary theory? A number of poets, including some of the greatest (e.g., Walt Whitman's "When I Heard the Learn'd Astronomer"), have held that nature is drained of wonder when we understand it scientifically. In Keats' words, a mere touch of "cold philosophy" can "clip an angel's wings."

This is a deep mistake. The basis for this error seems to be that the poets, like many philosophers, have seen emotion and intellect as distinct and opposed. The poets judged feeling to be better than intellect; the philosophers made the opposite judgment. In fact, of course, human nature is not so simplistically bifurcated, and feeling and intellect need not be in opposition. It is no accident that the greatest scientists were passionate thinkers who yearned for understanding as the mystic yearns for God.

Myth is not opposed to science unless we lapse into the follies of the fundamentalists and try to turn myth into science. It is precisely by taking myth as myth that we yield to its transformative power. The various neopagan movements each celebrates and expounds its own ancient mythos. To adopt such a tradition and make it your own is to put yourself into

a community that extends deep into prehistory, but which offers you a sense of wonder that is augmented, not diminished, by scientific understanding.

### 5. Conclusions

It appears, then, that there are identifiable modes of spirituality that were common among ancient pagans, and which can be recovered by today's neopagans. Like the ancients, we can regard religion not as a means of salvation or release from a dismal state, but as an affirmation of life's riches and values and a source of comfort and stability through life's inevitable travails and tragedies. Further, we can experience the world as the interweaving of the sacred and the secular, the transcendent and the mundane. We also can rediscover the power of myth: It connects us to our past, centers us in the present, and opens unexpected possibilities of feeling and imagination.

Of course, I have here only vaguely characterized these modes of spirituality; each branch of neopagans will express them in the idiom of their own traditions. There is no reason why core elements of ancient spirituality cannot be identified, recovered, adapted, and made a crucial part of our spiritual experience today, as the Icelandic Ásatrú Fellowship and many other neopagan groups have done. Naysayers cannot burden neopagans with requirements stricter than any that are met by other post-axial religions. If, despite the upheavals of twenty centuries, current worshippers can legitimately call themselves Christian, then there seems to be no greater difficulty in identifying as a pagan. Further the enormous changes in our material and intellectual environment over thousands of years do not preclude the existence of permanent possibilities of spirituality, and their potential to enrich current lives as much as ancient ones.

**Funding:** This research received no external funding.

**Conflicts of Interest:** The author declares no conflict of interest.

### Notes

[1]　The Axial Age, dated very roughly from 800 to 200 BCE, is the period religious scholars have identified as the era in which the archaic religions were replaced by what John Hick calls "the religions of salvation or liberation (Hick 2004, p. 29)." This was the period of Confucius and Lao Tzu in China, of The Buddha and Mahavira in India, of Zoroaster in Persia, and of the great Hebrew prophets. The term "Axial Age" derives from Karl Jaspers' identification of the *Achsenzeit,* the occurrence, geographically widespread but concentrated in time, when major religious and philosophical figures arose in diverse cultures.

[2]　Greek religion, for instance, was unified by a common body of myth, by sites universally recognized as of religious significance, such as Delphi, and by the celebration of panhellenic festivals such as the Olympic Games. However different *poleis* celebrated different festivals, such as the *Karneia* and *Hyacinthia* in Sparta and the *Heraea* in Argos (see Adkins and Adkins 1997, pp. 356–57). In fact, as the *Oxford Companion to Classical Civilization* notes:

Turning to the analysis of Greek religion as it appears in the post-Geometric period, we find in common with most pre-modern societies a strong link between religion and society to the extent that the sacred/secular dichotomy as we know it has little meaning for the Greek world. Greek religion is community based, and to the extent that the *polis* forms the most conspicuous of communities, it is therefore *polis*-based. (Hornblower and Spawforth 1998, p. 590)

There were, therefore, significant local variations in ritual and practice.
As for Norse religion, as David M. Wilson notes:

This was not a centrally organized religion; although there were cult places, temples and altars (some of which were more important than others), there was no strict religious discipline. The priests were not set apart . . . There was no recognized doctrine, no uniform method of worship; a man chose his own god and went his own way calling on different gods in different circumstances. (Wilson 1989, p. 42)

[3]　Mircea Eliade's *The Sacred and the Profane,* is, of course, the classic account of ancient religiosity with many examples of the ritualistic creation of sacred time and space.

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
