# Peer review of "Reviving Pagan Spirituality: A Manifesto"

_religions, doi:10.3390/rel13100942_

Round 1

Reviewer 1 Report

This sharp article is revelatory.  Accurately, to 'reclaim lost religiosity' -line 172  in the pre-axial religions which would take life As Is -line .229. 

(some post-axial heresies and variations may have wanted to restore the break from nature, the Saivites and saddhus and tantrikas, the Shakers and Cathars in some ways, Shinto Animism in Japan is a daily integration, or can be.  Emily Dickinson at her garden.  Pollock's 'i am nature' wasn't amiss.). 

line 263, the 'emergency character' of the post-axial is interesting.  Again, i'm delighted to navigate myself back to the Communitas (Turner) of my people i have gathered about me, today. 

line 148,  the symmetrical in religion off of culture, that we can flip it:

the neo pagan experience may regain a synergistic rapport with nature instead of the exploitive dominator model of the Hebrew genesis narrative. "Adam, god says you have sovereinty over the oceans over the plants and animals and, sure, why not, over Eve- and Asherah, poof."  Reindeer herders tender their animals, working together. That can happen again, imagine that for fishing our oceans today where the whales are happy. Too.  

To consider also: Spirit of Land is critical for paganism, what is our conversation with the land we are on at the ritual.  And when can we and must we invent the ritual on the spot, to function, to regain the experience for the person reclaiming religiosity at that moment. 

Channoya. Tea ceremony, works to that moment. Elders across the world invite one to tea.   The article is solid. Suggesting that life can be lived as is, creatively side stepping Anomie- line 225.  Emily Dickinson as our elder.     

Author Response

Thanks much for your input!

Reviewer 2 Report

I do not take issue with the author's definition of the 'Axial-age'. However, the origin of the term, Axial-age, is often traced back to Karl Jaspers, who defines it quite differently. I note that Jaspers is not cited in this paper. Some clarification on this point would be helpful.

Author Response

Thanks much for your input!